# A prospective study of the adaptive changes in the gut microbiome during standard-of-care chemoradiotherapy for gynecologic cancers

Molly B. El Alam[1], Travis T. Sims[2], Ramez Kouzy[1], Greyson W. G. Biegert[1], Joseph A. B. I. Jaoude[1], Tatiana V. Karpinets[3], Kyoko Yoshida-Court[1], Xiaogang Wu[3], Andrea Y. Delgado-Medrano[1], Melissa P. Mezzari[4], Nadim J. Ajami[3], Travis Solley[1], Mustapha Ahmed-Kaddar[1], Lilie L. Lin[1], Lois Ramondetta[2], Amir Jazaeri[2], Anuja Jhingran[1], Patricia J. Eifel[1], Kathleen M. Schmeler[2], Jennifer Wargo[5], Ann H. Klopp[1]*, Lauren E. Colbert[1]*

1 Department of Radiation Oncology, The University of Texas MD Anderson Cancer Center, Houston, TX, United States of America, 2 Department of Gynecologic Oncology and Reproductive Medicine, The University of Texas MD Anderson Cancer Center, Houston, TX, United States of America, 3 Department of Genomic Medicine, The University of Texas MD Anderson Cancer Center, Houston, TX, United States of America, 4 Department of Molecular Virology and Microbiology, Alkek Center for Metagenomics and Microbiome Research, Baylor College of Medicine, Houston, TX, United States of America, 5 Department of Surgical Oncology, The University of Texas MD Anderson Cancer Center, Houston, TX, United States of America

☯ These authors contributed equally to this work.

* aklopp@mdanderson.org (AHK); lcolbert@mdanderson.org (LEC)

**Data Availability Statement:** Data are deposited in Sequence Read Archive (SRA). The project accession number is PRJNA685389.

## Abstract

### Background

A diverse and abundant gut microbiome can improve cancer patients' treatment response; however, the effect of pelvic chemoradiotherapy (CRT) on gut diversity and composition is unclear. The purpose of this prospective study was to identify changes in the diversity and composition of the gut microbiome during and after pelvic CRT.

### Materials and methods

Rectal swabs from 58 women with cervical, vaginal, or vulvar cancer from two institutions were prospectively analyzed before CRT (baseline), during CRT (weeks 1, 3, and 5), and at first follow-up (week 12) using 16Sv4 rRNA gene sequencing of the V4 hypervariable region of the bacterial 16S rRNA marker gene. 42 of these patients received antibiotics during the study period. Observed operational taxonomic units (OTUs; representative of richness) and Shannon, Simpson, Inverse Simpson, and Fisher diversity indices were used to character-ize alpha (within-sample) diversity. Changes over time were assessed using a paired t-test, repeated measures ANOVA, and linear mixed modeling. Compositional changes in specific bacteria over time were evaluated using linear discriminant analysis effect size.

### Results

Gut microbiome richness and diversity levels continually decreased throughout CRT (mean Shannon diversity index, 2.52 vs. 2.91; all $P < 0.01$), but were at or near baseline levels in 60% of patients by week 12. Patients with higher gut diversity at baseline had the steepest

**Funding:** This research was supported in part by the Radiological Society of North America Resident/Fellow Award (to L.E.C.); by the National Institutes of Health (NIH) through MD Anderson's Cancer Center Support Grant P30CA016672 and through T32 grant CA101642-14 (T.T.S.); and by The University of Texas MD Anderson Cancer Center HPV-related Cancers Moonshot (L.E.C., A. K.). The funding sources were not involved in the development of the research hypothesis, study design, data analysis, or manuscript writing. Data access was limited to the authors of this manuscript.

**Competing interests:** The authors have declared that no competing interests exist.

decline in gut microbiome diversity. Gut microbiome composition was significantly altered during CRT, with increases in Proteobacteria and decreases in Clostridiales, but adapted after CRT, with increases in Bacteroides species.

## Conclusion

After CRT, the diversity of the gut microbiomes in this population tended to return to baseline levels by the 12 week follow-up period, but structure and composition remained significantly altered. These changes should be considered when designing studies to analyze the gut microbiome in patients who receive pelvic CRT for gynecologic cancers.

## Introduction

The current standard of care for patients with human papillomavirus (HPV)-related gynecologic cancers is concurrent chemoradiotherapy (CRT), which is generally followed by brachytherapy [1–3]. Despite such therapy, approximately 30–40% of patients develop recurrent disease.

Potentially relevant biomarkers for predicting treatment response in gynecologic cancer patients include tumor genomic markers, immune response markers, and tumor microenvironment markers. One intriguing tumor microenvironment marker is the microbiome, which may affect immune response during and after CRT. The microbiome is composed of various communities of bacteria, fungi, viruses, and archaea that coexist with their host [4,5]. Gut microbiota have been shown to affect host immunity, play a role in carcinogenesis, and influence therapy response [6–9]. At least one study has shown that more diverse and abundant gut microbiota can improve immunotherapy response [10]. Because HPV-related cancers are primarily mucosal and virally induced, the microbiome may play a significant role in these tumors' response to treatment. The gut is the largest immune organ in the body, and mucosa play a key role in regulating immune response to outside pathogens in conjunction. Determining the gut microbiome's role in influencing treatment response in gynecologic cancer patients requires an improved understanding of the ways in which CRT itself affects the gut microbiome.

The gut microbiota of cervical cancer patients and healthy individuals differ significantly [11], and increasing data suggest that the gut microbiome plays a role in gynecologic cancer including, but not limited to, treatment response and survival [12]. Like patients with gynecologic cancers, patients with other cancers, including anal, rectal, prostate, cervical, and endometrial cancer, are treated primarily with pelvic radiotherapy, with or without chemotherapy, and likely have similar temporal changes in the diversity and composition of the gut microbiome over time as a result of CRT; However, prospective data on the changes the gut microbiome undergoes during pelvic radiotherapy remain scarce, as most studies on the subject have provided only single time-point or retrospective data. Having prospective data is particularly important for studying the temporal changes in the gut microbiome. We hypothesize that CRT could induce structural and possibly compositional changes in the gut microbiome of patients with gynecologic cancers. The purpose of this study was to prospectively determine the effect of standard pelvic CRT on the gut microbiome in a cohort of women with locally advanced gynecologic cancers. Our findings could be used to guide future studies of the gut microbiome as a predictive and prognostic biomarker.

## Materials and methods

### Patients

Under a protocol approved by the UT M.D. Anderson Cancer Center Institutional Review Board (MDACC 2014–0543), we prospectively collected rectal swab specimens from 58 gynecologic cancer patients who received standard-of-care pelvic CRT at MD Anderson or the Lyndon B. Johnson Hospital Oncology Clinic from September 22, 2015, to January 11, 2019. Patients with newly diagnosed, biopsy-proven locally advanced cervical, vaginal, or vulvar carcinoma were eligible for the study. Our clinic sees an average of 75 patients per year. All eligible patients seen in consultation in the gynecologic radiation oncology clinic were approached by a member of the research team for consideration of the study. Patients who previously received pelvic radiation or systemic therapy were excluded. Informed written consent was obtained from 58 eligible patients willing to participate in our study. Eligible patients received CRT with external beam radiotherapy and weekly cisplatin, which was followed by brachytherapy if indicated. All patients received a minimum radiation dose of 45 Gy given in 25 fractions over 5 weeks along with cisplatin at 40 mg/m$^2$ and either 2 pulsed dose rate brachytherapy treatments or 5 high dose rate brachytherapy treatments. The majority of patients were tested for the Human Papillomavirus (HPV). Patients' antibiotic use during the study period was determined by searching inpatient and outpatient pharmacy prescription records. Demographic data including age, race/ethnicity, and smoking status were also collected from patients' medical records. Clinical characteristics including BMI, cancer type, stage, grade, histology, and node level were also obtained from patients' medical records. Our initial accrual goal was 60 patients in order to generate descriptive statistics for this pilot feasibility study.

### Sample collection, processing, and sequencing

Rectal swab samples were collected immediately before treatment initiation and 1, 3, 5, and 12 weeks after treatment initiation. Week 1 samples were collected only after patients received 1 −5 fractions of radiotherapy. We collected samples using quick-release matrix designed Isohelix DNA swabs. Lysis buffer (400 mL) was added to each sample within 1 hour of collection, and all samples were stored at -80˚C until they were subjected to RNA extraction and sequencing [12].

Sequencing of the V4 hypervariable region of the bacterial 16S rRNA marker gene (16Sv4) was performed. The 16Sv4 gene is both the most conserved and the most variable segment of the bacterial genome, making it the ideal target for performing phylogenetic analysis. 16S performed at the Alkek Center for Metagenomics and Microbiome Research at Baylor College of Medicine using a methodology from the Human Microbiome Project [13]. The sequencing was performed as described previously using strict OTU assignment criteria for rigorous classification [12]. Operational taxonomic unit (OTU) assignment was performed using clustering with 97% sequence similarity using in-house pipelines developed for the human microbiome project. The range and average sequence sampling depth for each sample is provided in S1 Table.

### Assessment of gut richness and diversity over time

We evaluated all available alpha (within-sample) diversity metrics, including richness (the absolute number of observed OTUs), Shannon diversity, Inverse Simpson diversity, Simpson diversity, and Fisher diversity. We used the paired Student t-test to compare the mean diversity metrics recorded throughout treatment to the mean baseline diversity metrics. To compare changes in gut richness and diversity across the first 4 time points for patients who provided

samples at those times (n = 23), we performed repeated measures analysis of variance (ANOVA) with Bonferroni correction for post hoc comparisons to identify timepoints that had different alpha diversities. For each patient who provided samples at baseline, week 5, and week 12 (n = 11), we tracked the changes in diversity over time to determine the number of patients for whom gut diversity returned to baseline levels. For patients who did not provide samples at all time points (n = 53), we built a linear mixed model for time as a predictor of diversity. Covariates were time, and an interaction term of baseline diversity and time, with individual patients as a random effect. To determine whether taking antibiotics affected the diversity of the gut microbiome, we collected information on patients' use of antibiotics prior to weeks 1, 3, and 5 and performed an independent Student t-test to compare the means of each diversity metric at each time point between patients who took antibiotics before that time point and patients who did not. We also performed an independent Student t-test to determine whether the fold change in diversity from baseline to week 12 was dependent on prior antibiotic use.

### Assessment of gut composition over time

We used linear discriminant analysis effect size [14] to examine beta (between-sample) diversity and identify changes in bacterial genera between baseline and week 5. To visualize the differences in gut microbiota between baseline and week 5, we calculated an OTU enrichment index, which is specifically designed to evaluate enrichment in the presence of rare species, as described previously [6]. The enrichment index can take values ranging from -1 (OTU found in week 5 but not baseline) to 1 (OTU found in baseline but not week 5). We generated a heatmap of OTU abundances, in which the columns represented samples and the rows represented the OTUs and used the overall abundance distribution of all OTUs to identify thresholds for low (green), medium (yellow), and red (high) abundances. Statistical significance was set at an alpha of 5% for a two-sided P-value. Analyses were conducted using RStudio 1.2.5033 [15].

## Results

### Patients

Patient characteristics are provided in S2 Table. Our patient population included 58 patients (55, 2, and 1 with cervical, vulvar, and vaginal cancer, respectively). The patients' mean age was 49.36 years (standard deviation [SD], 10.52 years). Only 4 patients received extended field radiation to the Para-Aortic (PA) nodes. Of the 58 patients, 42 took antibiotics during the study period, of whom 8 patients received antibiotics prior to week 1, 13 patients prior to week 3, and 18 patients prior to week 5 (S1 Fig). Twelve patients received only sulfamethoxazole trimethoprim with brachytherapy at week 5. Reasons for antibiotic use among our patients include urinary tract infections (UTI), skin superinfections, and use of a urinary foley catheter.

Most patients (79.3%) had squamous cell carcinoma. Of the 56 patients whose HPV status we determined, 82% were positive for the virus. Baseline samples were collected from 55 patients; week 1 samples, from 38 patients; week 3 samples, from 36 patients; week 5 samples, from 47 patients; and week 12 samples, only 16 patients (S3 Table).

### Gut richness and diversity over time

Overall, the gut richness and diversity metrics significantly decreased by week 5 but returned to baseline levels after CRT. The baseline means for observed OTUs and the Shannon, Simpson, Inverse Simpson, and Fisher diversity indices were 107.58 (SD, 34.60), 2.91 (SD, 0.59), 0.87 (SD, 0.09), 11.39 (SD, 6.55), and 18.26 (SD, 6.98), respectively (Table 1). None of the

**Table 1. Differences between baseline diversity metrics and those assessed during and after chemoradiotherapy.**

| Diversity metric | Time point | Mean ± SD | $P^a$ |
|---|---|---|---|
| Observed OTUs | Baseline | 107.58 ± 34.60 | - |
| | Week 1 | 101.74 ± 32.38 | 0.625 |
| | Week 3 | 99.08 ± 25.02 | 0.064 |
| | Week 5 | 83.79 ± 26.28 | **<0.001** |
| Shannon | Baseline | 2.91 ± 0.59 | - |
| | Week 1 | 2.92 ± 0.46 | 0.599 |
| | Week 3 | 2.93 ± 0.45 | 0.400 |
| | Week 5 | 2.52 ± 0.68 | **<0.001** |
| Simpson | Baseline | 0.87 ± 0.09 | - |
| | Week 1 | 0.89 ± 0.06 | 0.566 |
| | Week 3 | 0.89 ± 0.06 | 0.844 |
| | Week 5 | 0.81 ± 0.15 | **0.002** |
| Inverse Simpson | Baseline | 11.39 ± 6.55 | |
| | Week 1 | 10.63 ± 4.17 | 0.326 |
| | Week 3 | 11.75 ± 5.95 | 0.441 |
| | Week 5 | 8.23 ± 5.05 | **0.001** |
| Fisher | Baseline | 18.26 ± 6.98 | - |
| | Week 1 | 17.07 ± 6.5 | 0.637 |
| | Week 3 | 16.45 ± 5.00 | 0.053 |
| | Week 5 | 13.53 ± 4.95 | **<0.001** |

[a]Paired Student t-test.

alpha diversity metrics at week 1 or week 3 differed significantly from those at baseline (Table 1; $P > 0.05$ for all); However, the alpha diversity metrics of the week 5 samples were significantly lower than those of the baseline samples: the mean observed OTU value was 83.79 ($P < 0.001$); the mean Shannon diversity index, 2.52 ($P < 0.001$); the mean Simpson diversity index, 0.81 ($P = 0.002$); the mean Inverse Simpson diversity index, 8.23 ($P = 0.001$); and the mean Fisher diversity index, 13.53 ($P < 0.001$). The repeated measures ANOVA test showed, that the mean observed OTUs at week 5 (81.91) were significantly lower than those at baseline (108.39; $P = 0.007$), week 1 (mean, 103.91; $P = 0.048$), and week 3 (mean, 97.17; $P = 0.024$) (Fig 1A). Furthermore, Fisher diversity index was lower at week 5 than at baseline ($P = 0.008$), week 1 ($P = 0.048$), and week 3 ($P = 0.026$) (Fig 1C). Although the Shannon, Simpson, and Inverse Simpson diversity indices decreased between baseline and week 5, they did not differ significantly between week 5 and either week 1 or 3 (Fig 1B, 1D and 1E). We found no significant differences in diversity metrics between patients in our study population who took antibiotics prior to each time point and those who didn't. Diversity at week 12 was also compared to baseline, however we have not included these results in Table 1 due to a low patient number at this timepoint.

The alpha diversity metrics at week 12 were not statistically lower than those at baseline ($P > 0.05$ for all). Of the 11 patients who provided samples at baseline, week 5, and week 12 (Fig 2), 4 (36%) had higher diversity at week 12 than at baseline, 4 (36%) had increased diversity after CRT that did not return to baseline levels, and 3 (27%) had a steady decrease in their microbiome diversity over time until follow up. We found no association between antibiotic use and gut microbiome diversity in our patient population.

Linear mixed modeling accounting for missing samples from baseline to week 5 (Table 2) both confirmed the above findings ($P < 0.01$) and showed that diversity decreased significantly

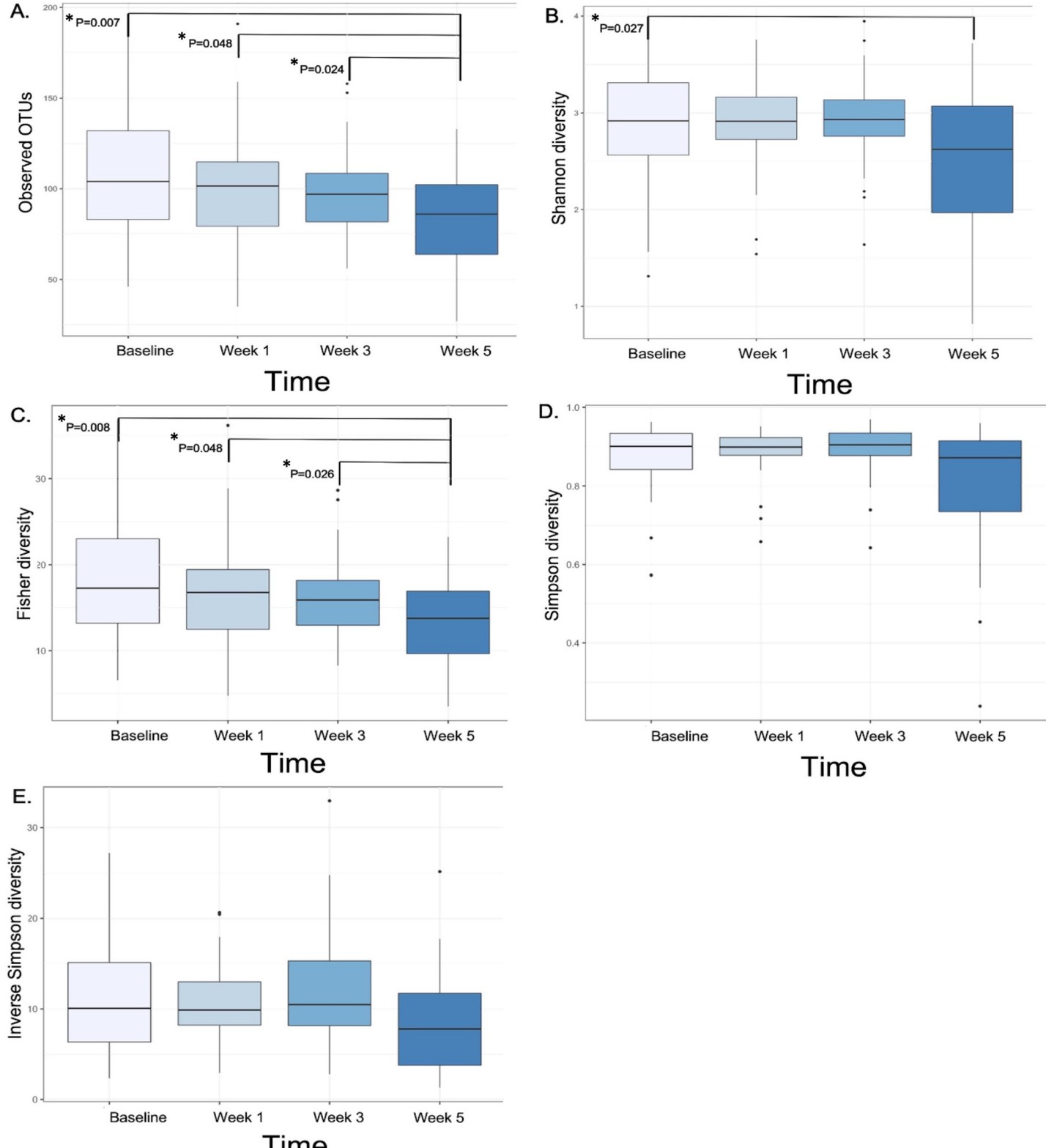

**Fig 1. Gut richness and diversity decrease significantly by week 5 of chemoradiotherapy.** Box plots show changes in (A) observed OTUs (richness) and in (B) Shannon, (C) Fisher, (D) Simpson, and (E) Inverse Simpson diversity indices over time as determined with repeated measures ANOVA. P-values >0.05 are not shown.

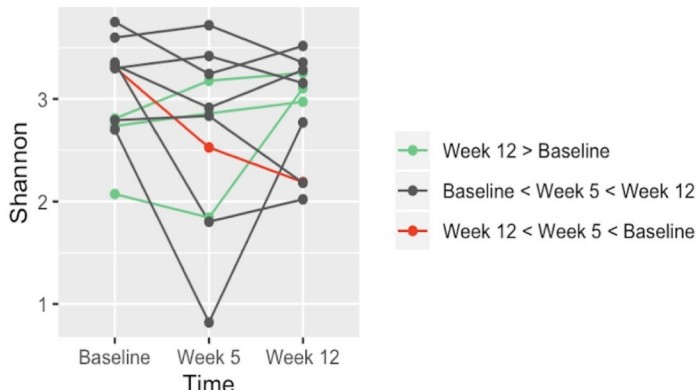

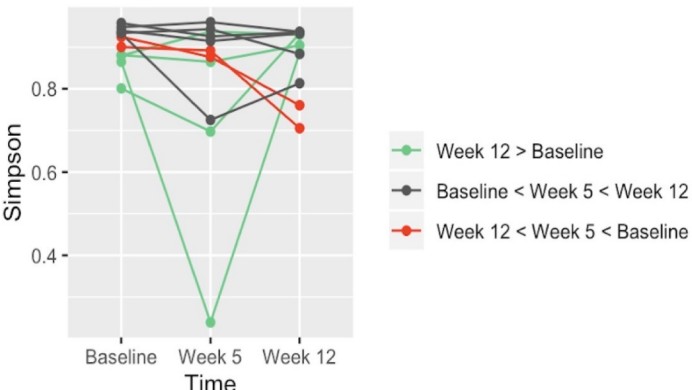

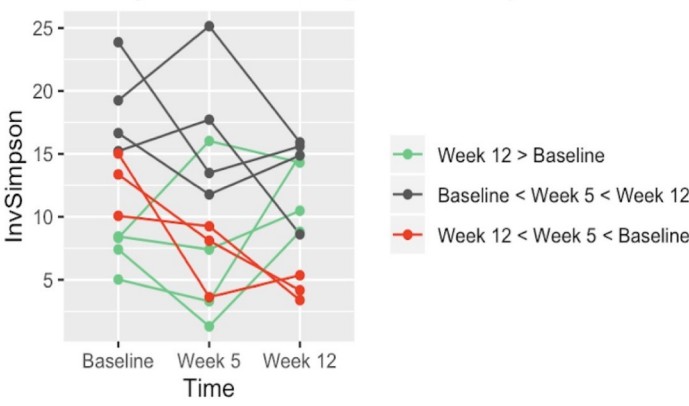

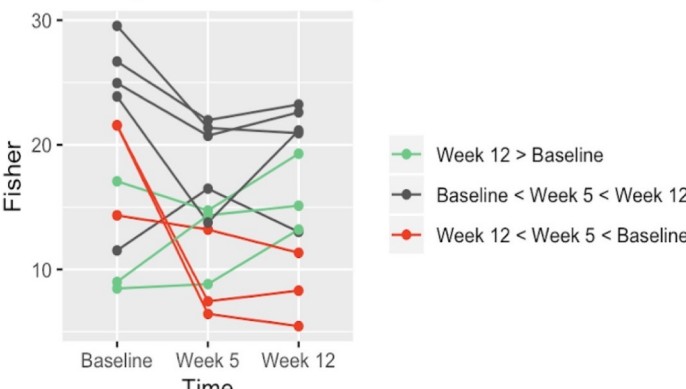

**Fig 2. Gut richness and diversity are at or near baseline levels by first follow-up after chemoradiotherapy.** Individual richness and diversity values are plotted for all patients who provided samples at baseline, week 5, and week 12. Each line denotes an individual patient. Most patients' gut diversity and richness returned or nearly returned to baseline levels. Green represents patients whose week 12 diversity levels were equal to or greater than baseline diversity levels. Black represents patients whose diversity levels decreased by week 5 but increased to near-baseline levels by week 12. Red represents patients whose diversity levels declined continuously from baseline to 12 weeks.

with time. In addition, for nearly all diversity indices, there was an interaction effect between baseline diversity and time ($P < 0.01$), indicating that baseline diversity is important in predicting the response of the gut microbiome to CRT (Fig 3). Patients with high baseline diversity

**Table 2. Results of the linear mixed model analysis of time as a predictor of diversity, accounting for patients with missing data for one or more time points[a].**

| Outcome | Covariate | Estimate | 95% confidence interval | P |
|---|---|---|---|---|
| Observed OTUs | Low Diversity | -51.106 | (-61.79, -40.42) | <0.01 |
| | Time | -7.569 | (-9.66, -5.47) | <0.01 |
| | Low Diversity:Time | 6.403 | (3.36, 9.48) | <0.01 |
| Shannon | Low Diversity | -0.80802 | (-1.03, -0.59) | <0.01 |
| | Time | -0.1181 | (-0.16, -0.07) | <0.01 |
| | Low Diversity:Time | 0.08781 | (0.02, 0.15) | <0.01 |
| Simpson | Low Diversity | -0.091656 | (-0.13, -0.05) | <0.01 |
| | Time | -0.013607 | (-0.02, 0) | <0.01 |
| | Low Diversity:Time | 0.002634 | (-0.01, 0.02) | 0.684 |
| Inverse Simpson | Low Diversity | -8.1326 | (-10.32, -5.95) | <0.01 |
| | Time | -1.1006 | (-1.56, -0.64) | <0.01 |
| | Low Diversity:Time | 1.1159 | (0.45, 1.78) | <0.01 |
| Fisher | Low Diversity | -10.5757 | (-12.66, -8.49) | <0.01 |
| | Time | -1.6527 | (-2.08, -1.22) | <0.01 |
| | Low Diversity:Time | 1.453 | (0.86, 2.05) | <0.01 |

[a] Samples collected 12 weeks after treatment initiation were excluded from the analysis.

metrics had steeper declines in their gut diversity over time than those with low baseline diversity metrics did.

## Gut composition over time

Using linear discriminant analysis effect size, we identified compositional changes in patients' gut microbiomes based on specific bacterial phyla, classes, orders, families, and genera that were differentially enriched in patients between baseline and week 5 (Fig 4A and 4B), between week 5 and week 12 (Fig 4C), and between baseline and week 12 (Fig 4D) (false discovery rate $P < 0.05$; linear discriminant analysis score >4). During CRT, the relative abundances of Clostridia, Clostridiales, *Faecalibacterium*, and *Ezakiella* decreased, whereas the relative abundances of Proteobacteria, Gammaproteobacteria, Bacilli, Pasteurellales, Pasteurellaceae, and *Haemophilus* increased (S2 Fig). Between week 5 and week 12, the abundances of Proteobacteria and Gammaproteobacteria continued to decrease, whereas that of Ezakiella returned to its baseline level. Still, we saw significant alterations in relative abundances of bacteria, primarily Bacteroidetes, between baseline and week 12 (S3 Fig).

Overall, the most significant alterations in composition during CRT were an increase in mostly pathogenic [16], gram-negative Proteobacteria (Fig 5A) and a decrease in generally beneficial Clostridia (Fig 5B), both of which tended to return to baseline levels after CRT. Thus, the most significant long-term alteration was a decrease in Bacteroidetes (Fig 5C), which likely helped prevent pathogenic bacteria from colonizing the gut and was a beneficial adaptation in response to a significant assault on the gut microbiome.

Rare species enrichment analysis (Fig 5D–5F) demonstrated an adaptation of rare species at the OTU level. The relative abundance of Clostridia as a class significantly decreased from baseline to week 5, whereas the pattern of changes of rare individual OTUs were more complex. Most levels of rare Clostridia species were significantly higher at baseline than at week 5. However, a small fraction of individual OTUs of Clostridia, increased their occupancy during CRT (Fig 5E).

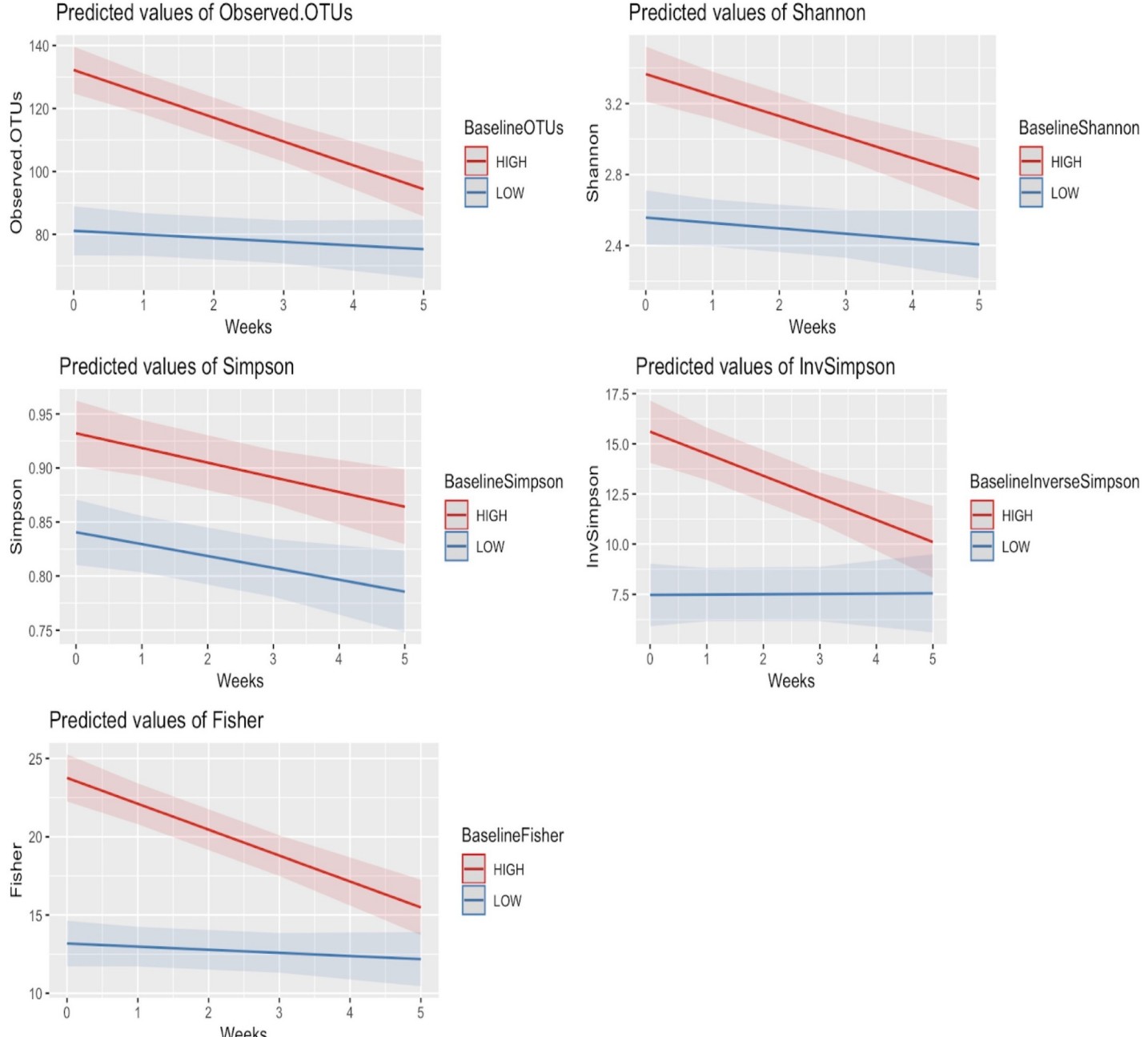

**Fig 3. Patients with high baseline gut richness and diversity have significantly greater decreases in diversity over time.** Linear mixed modeling demonstrated an interaction between time and baseline diversity values for all metrics except Simpson diversity. Patients with high baseline diversity (red) had a significantly steeper decline in diversity than patients with low baseline diversity (blue) did.

## Discussion

### Principal findings

We hypothesized that the process of undergoing standard-of-care CRT induces compositional and/or functional changes in the gut microbiome that are relevant for analysis of the gut microbiome as a future biomarker. Our prospective longitudinal analysis showed that gut microbial diversity is stable immediately after the initiation of CRT but declines significantly

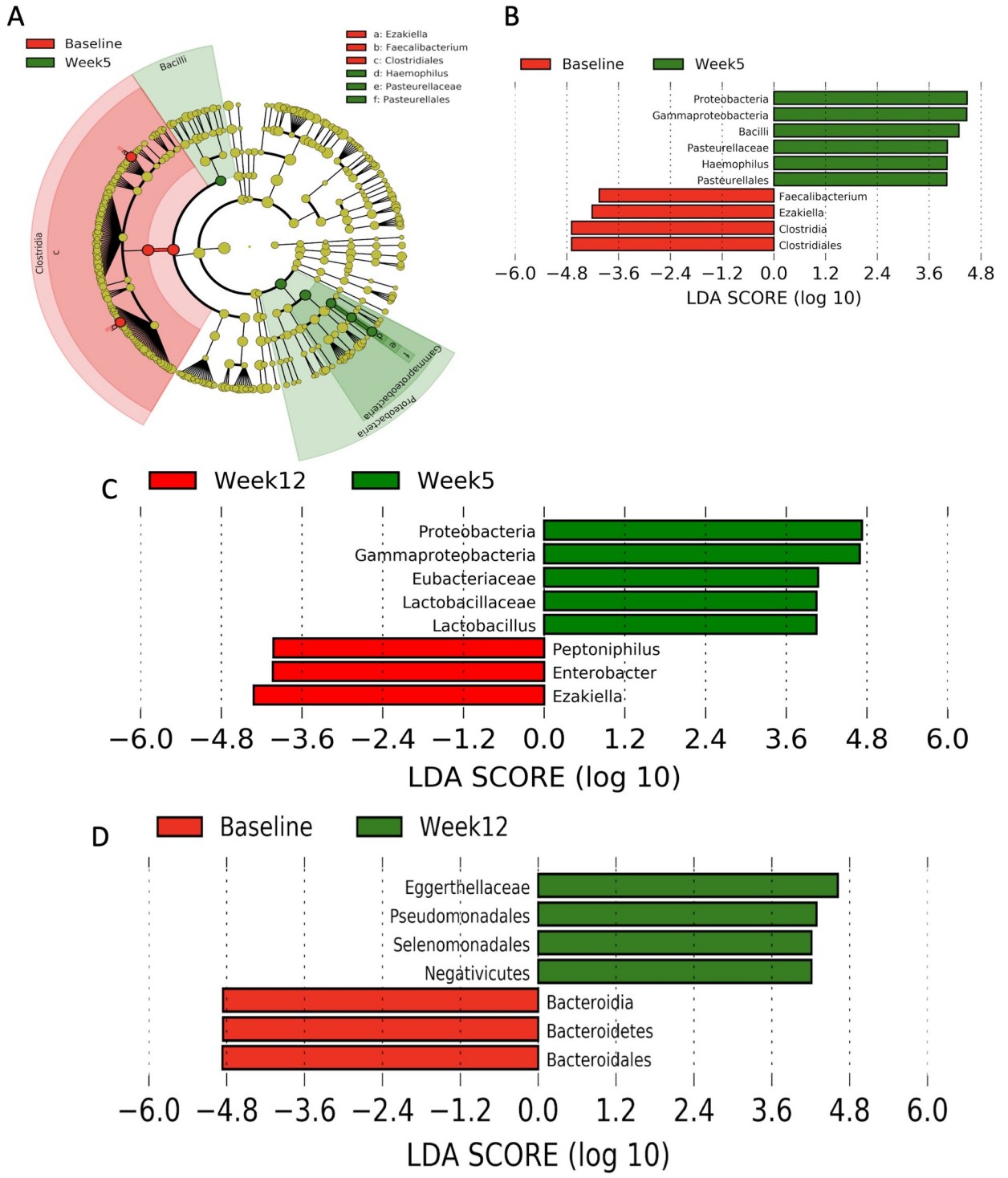

**Fig 4. Gut composition changes significantly during chemoradiotherapy, adapts after treatment, and remains altered at first follow-up.** The differential enrichments of bacterial phyla, classes, orders, families, and genera between baseline and week 5 were identified by linear discriminant analysis effect size. Significant differences are those with a logarithmic linear discriminant analysis score >4 and a factorial Kruskal-Wallis p-value <0.05. Bacteria that were most significantly altered during chemoradiotherapy (between baseline and week 5; A and B) included the Clostridia class and Proteobacteria phylum. After chemoradiotherapy (between week 5 and week 12; C), proteobacteria decreased, whereas Ezakiella increased. Overall, the most significant alteration between baseline and first chemoradiotherapy follow-up (week 12; D) was an increase in Bacteroidales.

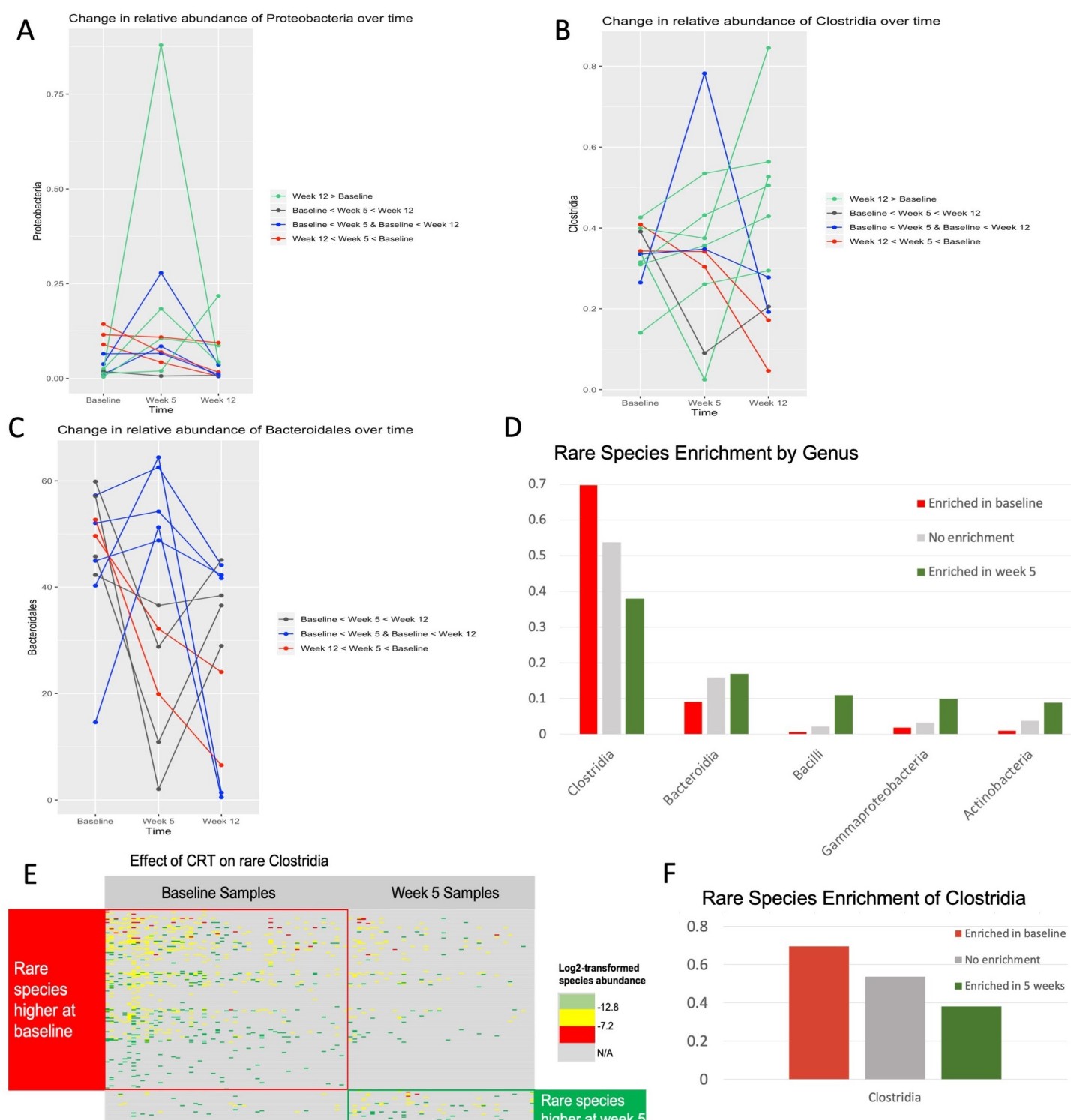

**Fig 5. Significant adaptations in gut composition occur during and after chemoradiotherapy.** The relative abundances of the pathogenic, gram-negative Proteobacteria phylum (A) increased significantly, whereas those of the beneficial members of the Clostridiales class (B) decreased significantly, during chemoradiotherapy but tended to return or nearly return to baseline levels by first follow-up (week 12). The most significant alteration between baseline and week 12 was an increase in the generally beneficial Bacteroidetes (C), which are known to help prevent pathogenic bacteria from colonizing the human gut and may be a sign of adaptation in response to increases in pathogenic bacteria during chemoradiotherapy. Enrichment analysis of rare species (D) suggests adaptation of individual microbial species at the OTU level. For example, individual species of Clostridia (E) were either enriched or depleted at week 5 as compared with baseline.

over the first 5 weeks of treatment. After patients complete CRT, their gut diversity variably returns to baseline levels, but gut composition and structure may remain altered. These findings are relevant to future studies of the gut microbiome during and after CRT.

Our findings that alterations in microbial composition and relative abundance occur within the first 5 weeks of CRT initiation and after CRT completion corroborate previous animal studies [17]. Furthermore, our study builds upon data reported by Nam et al. [18], who established that pelvic radiotherapy alters the composition of the gut microbiome of gynecologic cancer patients.

Our findings also provide additional insight into the temporal changes of the microbiome during CRT. The temporal treatment-induced shift in the gut microbiome that we observed is a unique effect that may influence the clinical outcomes of gynecologic cancer patients [19]. Identifying the mechanism by which this shift occurs was beyond the scope of the current study, but this mechanism could be related to changes in the gut epithelium and mucus layer that are followed by the overgrowth of pathogenic bacteria, which could in turn affect immune cells' maturation and responses to tumor. Cumulative fractions of CRT may induce the outgrowth of these radioresistant or pathologic microbial taxa, resulting in the selection of taxa that are tolerant to radiation-induced insults.

Our findings show that baseline gut diversity is a strong predictor of the degree of change in gut diversity during CRT. Compared with patients with low gut diversity at baseline, those with high gut diversity at baseline had a greater decline in gut diversity from baseline to week 5. This was an unexpected finding because gut diversity is commonly seen as a marker of vulnerability and immunogenicity [6,20], and high gut diversity is thought to be beneficial. We expected to see an overall decline in gut diversity at 5 weeks in all patients, regardless of their baseline gut diversity, but this finding suggests that the best target group for intervention may actually be patients with high baseline gut richness and diversity, rather than those with low baseline richness and diversity. We did not specifically investigate these high-diversity patients' gut composition in the present study, and larger studies are warranted to investigate the impact of gut composition on maintaining diversity.

CRT significantly impacted gut microbial composition in this longitudinal analysis. Strong adaptations of microbiota composition after CRT were observed in almost every patient. The gut microbiome is intrinsically dynamic [18], and this should be taken into account when considering how CRT impacts its composition over time. Here we show that the gut microbiome shifts from its baseline state towards a new composition that is represented by the relative abundance of different taxa. Specifically, we found that a variety of bacterial species, namely Clostridiales and Proteobacter, were significantly altered during CRT but returned to baseline levels following the completion of the therapy. Other species, namely Bacteroidetes, remained altered during our 12 week follow up period. Interestingly, preliminary investigations have shown that the relative abundance of Clostridia may serve as a potential marker for human health and cancer treatment response [21,22].

Our longitudinal data demonstrate the variability of the gut microbiome during CRT; as both pathogenic and beneficial organisms respond to the shifting microenvironment, the abundance of certain taxa expands while that of others declines. Importantly, the bacterial community 3 months after treatment remained altered as compared with that at baseline, indicating that the gut microbiome of the patients who completed CRT did not fully recover over this time, despite possible oncologic clinical recovery. This appears to be due to a prevalence of pathogenic bacteria and a decrease of beneficial bacteria during CRT, followed by an adaptive shift in Bacteroidetes to overgrow these pathogenic bacteria and allow beneficial Clostridiales to return to baseline levels. Others have reported similar findings in patients undergoing fecal microbiota transplantation [22,23]. In our analysis, we accounted for the effect of antibiotic

use on the gut microbial profiles, but we were unable to account for the effects of patients' chemotherapy, other medications, or diet, which should be considered in future studies. However, our findings support the notion that in some patients, certain therapeutic interventions can lead to the permanent divergence of the gut microbiome.

## Clinical implications

Our findings corroborate limited existing research showing that CRT diminishes the gut microbiome in cancer patients. Up to 20% of patients who have had CRT have long-term diarrhea after CRT [24], and it is possible these changes are related to lack of re-establishment of gut microbiome. We have previously shown that increased short term radiation toxicity is associated with a decreased gut microbiome [12]. Many studies have sought to examine this relationship between the intestinal effects of radiotherapy and the gut microbiome but have mostly established an association rather than a causation in any one direction [25,26]. Future studies should examine long-term gut toxicity for these patients. This study was not powered to detect effects of antibiotics on the gut microbiome, but we did not find an association between the administration of antibiotics for treatment-related toxicities and alteration of the gut microbiome in this patient population.

Modifying the gut microbiome to accumulate CRT-tolerant species could potentially be used to reduce treatment toxicity. Researchers in diverse areas of medicine have studied the treatment-enhancing and toxicity-limiting utility of the gut microbiome [17,20]. In one murine model, radiation-induced dysbiosis increased susceptibility to radiotherapy-related gastrointestinal toxic effects [27]. One such effect, chronic radiation enteritis, rarely occurs after pelvic CRT but can severely affect quality of life. Although epithelial disruption and damage certainly play a role in radiation enteritis, a paucity of "good bacteria," such as Clostridiales, might also potentiate chronic enteritis. One recent study suggested that fecal microbiota transplantation might improve severe radiation enteritis, which raises the possibility that replacing bacterial populations with more normally balanced populations could be curative [28]. Wang et al. [26] recently reported the first case series of patients with immune checkpoint inhibitor−associated colitis successfully treated with fecal microbiota transplantation. We have also previously witnessed the effect of low microbiome diversity on increased acute radiation toxicity [13]. Regardless of these findings, determining whether changes in the human gut microbiome during CRT affect patients' risk of treatment-related toxic effects warrants further investigation.

## Strengths and limitations

The present study had some limitations. First, because this study included patients at two institutions within the greater Houston area, its results might not be generalizable to a broader geographic population. Still, our study is one of the largest longitudinal analyses of changes in the gut microbiome in patients with gynecologic malignancies. Second, owing to the longitudinal study design, some of the patients were inevitably lost to follow-up. The only requirement to be included in this study was the collection of a baseline sample, which might explain this high attrition rate. Patients were offered the opportunity to give samples at each follow-up time point but were not required to do so. Moreover, we were unable to collect samples at certain time points if patients were in pain. Longitudinal sampling remains one of the most robust approaches to analyzing microbiome data, and thus we collected as many serial samples as possible on study. We also followed a set protocol from sample collection through 16S sequencing and controlled for batch effects with each sequencing run in order to limit artifactual variations. Patients undergoing radiation for cervical cancer are at a high risk of urinary

tract infections and skin infections. Many patients have advanced tumors which require nephrostomy tubes and/or indwelling urinary catheters which put them at a higher risk of infection. Thus, due to the high usage of antibiotics among our patients we did not have adequate power to identify differences in the gut microbiome between those who took antibiotics and those who did not. However, our results highlight the importance of continued research on this understudied topic.

Data from this study will be used to help build dynamic predictive models for treatment response and prognosis and guide future studies of interventions for chronic radiation enteritis in the case of gynecologic malignancies. The serial changes in the gut microbiome during CRT we present should be considered when designing studies aimed at analyzing the effects of the gut microbiome on treatment response and toxicity.

## Supporting information

**S1 Fig. Cumulative patients receiving antibiotics.** Number of patients who took antibiotics prior to each time point are presented. Total number of patients receiving antibiotics during the study period is 42.
(TIF)

**S2 Fig. Changes in relative abundances of different species between baseline and week 5.** Relative abundances of species identified using linear discriminant analysis effect size as having changed between baseline and week 5. Whiskers on the plot represent the confidence interval.
(TIF)

**S3 Fig. Changes in relative abundances of species between baseline and week 12.** Relative abundances of species identified using linear discriminant analysis effect size as having changed between baseline and week 5 are plotted. Whiskers on the plot represent the confidence interval.
(TIF)

**S1 Table. Range and average sequence sampling depth for each sample included in the study.**
(DOCX)

**S2 Table. Patient demographic and clinical characteristics (N = 58).**
(DOCX)

**S3 Table. Samples available for each patient at each time point.**
(DOCX)

## Acknowledgments

We thank the patients who participated in the study. We would like to acknowledge Mr. Joe Munch on his editorial support on this manuscript.

## Author Contributions

**Conceptualization:** Molly B. El Alam, Travis T. Sims, Lilie L. Lin, Lois Ramondetta, Amir Jazaeri, Anuja Jhingran, Patricia J. Eifel, Kathleen M. Schmeler, Jennifer Wargo, Ann H. Klopp, Lauren E. Colbert.

**Data curation:** Molly B. El Alam, Lilie L. Lin, Lois Ramondetta, Amir Jazaeri, Anuja Jhingran, Patricia J. Eifel, Kathleen M. Schmeler, Jennifer Wargo, Ann H. Klopp, Lauren E. Colbert.

**Formal analysis:** Molly B. El Alam, Travis T. Sims, Ramez Kouzy, Greyson W. G. Biegert, Joseph A. B. I. Jaoude, Tatiana V. Karpinets, Kyoko Yoshida-Court, Xiaogang Wu, Andrea Y. Delgado-Medrano, Melissa P. Mezzari, Nadim J. Ajami, Travis Solley, Ann H. Klopp, Lauren E. Colbert.

**Funding acquisition:** Ann H. Klopp, Lauren E. Colbert.

**Investigation:** Molly B. El Alam, Ann H. Klopp, Lauren E. Colbert.

**Methodology:** Travis Solley, Ann H. Klopp, Lauren E. Colbert.

**Software:** Molly B. El Alam.

**Supervision:** Ann H. Klopp, Lauren E. Colbert.

**Visualization:** Molly B. El Alam, Travis T. Sims, Greyson W. G. Biegert, Ann H. Klopp, Lauren E. Colbert.

**Writing – original draft:** Molly B. El Alam, Travis T. Sims, Ramez Kouzy, Greyson W. G. Biegert, Joseph A. B. I. Jaoude, Tatiana V. Karpinets, Kyoko Yoshida-Court, Xiaogang Wu, Andrea Y. Delgado-Medrano, Melissa P. Mezzari, Nadim J. Ajami, Travis Solley, Mustapha Ahmed-Kaddar, Lilie L. Lin, Lois Ramondetta, Amir Jazaeri, Anuja Jhingran, Patricia J. Eifel, Kathleen M. Schmeler, Jennifer Wargo, Ann H. Klopp, Lauren E. Colbert.

**Writing – review & editing:** Molly B. El Alam, Travis T. Sims, Ramez Kouzy, Greyson W. G. Biegert, Joseph A. B. I. Jaoude, Tatiana V. Karpinets, Kyoko Yoshida-Court, Xiaogang Wu, Andrea Y. Delgado-Medrano, Melissa P. Mezzari, Nadim J. Ajami, Travis Solley, Mustapha Ahmed-Kaddar, Lilie L. Lin, Lois Ramondetta, Amir Jazaeri, Anuja Jhingran, Patricia J. Eifel, Kathleen M. Schmeler, Jennifer Wargo, Ann H. Klopp, Lauren E. Colbert.

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
