## [Decision Letter · Decision Letter 0]

8 Oct 2020

PONE-D-20-16939

Adaptive changes in the gut microbiome during standard-of-care chemoradiotherapy for gynecologic cancers

PLOS ONE

Dear Dr. Colbert,

Thank you for submitting your manuscript to PLOS ONE. After careful consideration, we feel that it has merit but does not fully meet PLOS ONE’s publication criteria as it currently stands. Therefore, we invite you to submit a revised version of the manuscript that addresses the points raised during the review process.

We had difficulty finding multiple reviewers for this manuscript. Based on my evaluation and that of one reviewer, we feel that there needs to be more concordance between the results and discussion. Some clarification and consolidation of the discussion is recommended as described in the reviewer's comments.

We look forward to receiving your revised manuscript.

Kind regards,

Christopher Staley, Ph.D.

Academic Editor

PLOS ONE

Journal Requirements:

2. Thank you for including your ethics statement: 'Under an Institutional Review Board-approved protocol (MDACC 2014-0543), we prospectively collected rectal swab specimens from 58 gynecologic cancer patients who received standard-of-care pelvic CRT at MD Anderson or the Lyndon B. Johnson Hospital Oncology Clinic between 2015 and 2019.'

(a) Please amend your current ethics statement to include the full name of the ethics committee/institutional review board(s) that approved your specific study.  

(b) Once you have amended this/these statement(s) in the Methods section of the manuscript, please add the same text to the “Ethics Statement” field of the submission form (via “Edit Submission”).

4. In your Methods section, please provide additional information about the participant recruitment method and the demographic details of your participants. Please ensure you have provided sufficient details to replicate the analyses such as: a) the recruitment date range (month and year) and b) a description of how participants were recruited.

5. Please provide a sample size and power calculation in the Methods, or discuss the reasons for not performing one before study initiation.

6.We note that you have indicated that data from this study are available upon request. PLOS only allows data to be available upon request if there are legal or ethical restrictions on sharing data publicly. For information on unacceptable data access restrictions, please see http://journals.plos.org/plosone/s/data-availability#loc-unacceptable-data-access-restrictions.

Reviewers' comments:

Reviewer's Responses to Questions

**Comments to the Author**

1. Is the manuscript technically sound, and do the data support the conclusions?

Reviewer #1: Partly

2. Has the statistical analysis been performed appropriately and rigorously? 

Reviewer #1: I Don't Know

3. Have the authors made all data underlying the findings in their manuscript fully available?

Reviewer #1: Yes

4. Is the manuscript presented in an intelligible fashion and written in standard English?

Reviewer #1: Yes

5. Review Comments to the Author

Reviewer #1: The authors conducted a prospective study to identify changes in the diversity and composition of the gut microbiome during and after pelvic CRT. Fifty-eight women with cervical, vaginal, or vulvar cancer from two institutions were prospectively analyzed before CRT (baseline), during CRT (weeks 1, 3, and 5), and at first follow-up (week 12). I congratulate them on completing their study and providing preliminary information regarding the effect of CRT on gut microbiome diversity. There is important information presented, however, the discussion/conclusions need to focus on the findings. In some cases the interpretation of the results seem far reaching.

Abstract:

Conclusion regarding "After CRT, the gut microbiome’s diversity tends to return to baseline levels,

but its structure and composition remain significantly altered." Seems excessive given the follow up period of only 7 weeks after radiation. Moreover, the authors state in the manuscript "Overall, two-thirds of patients’ diversity indices returned to near-baseline levels". The conclusion needs to be revised to communicate that there was short term follow up and include info regarding return to near-baseline levels. Further evaluation will be needed to determine if these changes persist and confirm the findings.

I agree with the following "These changes should be considered when designing studies to analyze the gut microbiome" but suspect that these temporal changes need to be considered for any study evaluating the gut microbiome in patient receiving chemoradiation and are not specific to only predictive/prognostic marker studies. The "predictive or prognostic biomarker . . ." phrase seems to come of out nowhere. In the discussion, "treatment response and toxicity" are mentioned.

Results:

Please include information regarding extended field radiation to the PA nodes.

Provide additional information regarding the reason for antibiotic use.

The authors "found no significant differences in diversity metrics between patients who took antibiotics prior to each time point and those who didn’t." However, with such a high number of patients receiving antibiotics at various time points there most likely was not adequate power to assess this question. Only 16 patients did not take antibiotics.

Discussion:

Further explain the high attrition rate - how does this compare to other studies of the gut microbiome?

Provide explanation for high antibiotic use.

Lines 308-310 - why do the authors believe a shift in gut microbiome with CRT will influence outcome? Add reference. If no reference revise this sentence

Line 337 How do you know Bacteroidetes, remained permanently altered when you only have short term follow up? "permanently altered" needs to be revised to reflect the period of time in your study

Line 359-361 - The authors state "Up to 20% of patients who have had CRT have long-term diarrhea after CRT

, and it is possible these changes are related to lack of re-establishment of gut microbiome." Could it also be possible that radiation therapy may effect stool frequency and bile salt absorption and that leads to the change in gut microbiota? Your study provides important preliminary information, however, there is still so much to learn about the long-term intestinal effects from radiation therapy and the relationship to the gut microbiome. It would be helpful to add information regarding long term damage to the intestinal mucosa from radiation if available in the literature.

Lines 362-364 - revise this sentences to reflect limited power in your study to determine association between antibiotics and the change in the gut microbiome. If you do have adequate power then please add this information

Lines 366-368 It is not clear how your study showed that intentionally modifying the gut microbiota could be used to reduce toxicity. Your results indicate a temporal change in microbiota composition in patients undergoing chemoradiation. Materials and methods don't describe an intervention to modify the gut microbiome; and results don't include toxicity data.

The discussion is too long and could be better organized.

6. PLOS authors have the option to publish the peer review history of their article (what does this mean?). If published, this will include your full peer review and any attached files.

Reviewer #1: No

---

## [Author Response · Author response to Decision Letter 0]

16 Dec 2020

Manuscript ID: PONE-D-20-16939

Title: Adaptive changes in the gut microbiome during standard-of-care chemoradiotherapy for gynecologic cancers

Dear PLOS ONE Editorial Team,

We are very excited to have been given the opportunity to revise our manuscript entitled “Adaptive changes in the gut microbiome during standard-of-care chemoradiotherapy for gynecologic cancers” for PLOS ONE. We carefully considered the comments offered by the reviewer. Herein, we explain how we revised the paper based on those comments and recommendations. We want to extend our appreciation for taking the time and effort necessary to provide such insightful guidance.

Note: Page and line numbers refer to the edited manuscript with revisions

EDITOR, COMMENT 1 

Include the full name of the ethics committee/institutional review board(s) that approved your specific study. 

please add the same text to the “Ethics Statement” field of the submission form (via “Edit Submission”)

- We appreciate the Editor’s comment. We have made the necessary addition to the methods section. The manuscript now reads “Under a protocol approved by the UT M.D. Anderson Cancer Center Institutional Review Board (MDACC 2014-0543)”

- Edit to the manuscript can be found on page 6, lines 79-80.

EDITOR, COMMENT 2 

 Please provide additional details regarding participant consent. In the ethics statement in the Methods and online submission information, please ensure that you have specified (1) whether consent was informed and (2) what type you obtained (for instance, written or verbal, and if verbal, how it was documented and witnessed). If your study included minors, state whether you obtained consent from parents or guardians. If the need for consent was waived by the ethics committee, please include this information.

- We appreciate the Editor’s comment. In our study, written consent was obtained and no minors were included. We have added the following statement in the methods section: “Informed written consent was obtained from 58 eligible patients willing to participate in our study.”

- Edit to the manuscript can be found on page 7, lines 88-89.

EDITOR, COMMENT 3 

In your Methods section, please provide additional information about the participant recruitment method and the demographic details of your participants. Please ensure you have provided sufficient details to replicate the analyses such as: a) the recruitment date range (month and year) and b) a description of how participants were recruited.

- We thank the Editor for their feedback. We have added the recruitment date range in more detail in the methods section: “from September 22, 2015, to January 11, 2019”. We have also added the following section on recruitment of patients: “All eligible patients seen in consultation in the gynecologic radiation oncology clinic were approached by a member of the research team for consideration of the study.”

- Details on the collection of demographic data have also been added to the methods section “Demographic data including age, race/ethnicity, and smoking status were collected from patients’ medical records. Clinical characteristics including BMI, cancer type, stage, grade, histology, and node level were also obtained from patients’ medical records”

- Edits to the manuscript can be found on page 6, line 83 and 85-87; page 7, lines 96-99. 

EDITOR, COMMENT 4 

Please provide a sample size and power calculation in the Methods, or discuss the reasons for not performing one before study initiation.

- We appreciate the editor’s comment. We did not conduct a power calculation prior to the initiation of this pilot study due to lack of prior data on the subject on which to base a power calculation. Instead, the accrual was planned for an initial 60 patients with a plan for generating descriptive statistics. 60 patients were accrued, but 2 patients were taken off study due to declining samples after initial consent. . We have added: “Our initial accrual goal was 60 patients in order to generate descriptive statistics for this pilot feasibility study.” to the methods section. We have also added our sample size to the following sentence. “We prospectively collected rectal swab specimens from 58 gynecologic cancer patients who received standard-of-care pelvic CRT at MD Anderson or the Lyndon B. Johnson Hospital Oncology Clinic from September 22, 2015, to January 11, 2019.”

- Edits to the manuscript can be found on page 7, lines 99-101; page 6, lines 80-85. 

REVIEWER 1, COMMENT 1 

The authors conducted a prospective study to identify changes in the diversity and composition of the gut microbiome during and after pelvic CRT. Fifty-eight women with cervical, vaginal, or vulvar cancer from two institutions were prospectively analyzed before CRT (baseline), during CRT (weeks 1, 3, and 5), and at first follow-up (week 12). I congratulate them on completing their study and providing preliminary information regarding the effect of CRT on gut microbiome diversity. There is important information presented, however, the discussion/conclusions need to focus on the findings. In some cases the interpretation of the results seem far reaching.

- We appreciate the reviewer’s comments. We believe that the significance of our study lies in establishing a foundation for what actually changes in the microbiome during CRT in patients receiving pelvic radiation to set the stage for future research. We have carefully reviewed our manuscript and have made necessary changes so as to not offer any conclusions that could be misinterpreted. 

Edits to the manuscript can be found on page 4, lines 30-32; page 11, lines 197-199; page 13, lines 213-215; pages 18-19, lines 314-317; page 20, lines 352-353; page 21, lines 383-386; page 22, lines 388-389; page 24, lines 426-429

REVIEWER 1, COMMENT 1 (Abstract) 

Conclusion regarding "After CRT, the gut microbiome’s diversity tends to return to baseline levels, but its structure and composition remain significantly altered." Seems excessive given the follow up period of only 7 weeks after radiation. Moreover, the authors state in the manuscript "Overall, two-thirds of patients’ diversity indices returned to near-baseline levels". The conclusion needs to be revised to communicate that there was short term follow up and include info regarding return to near-baseline levels. Further evaluation will be needed to determine if these changes persist and confirm the findings.

- We appreciate the reviewer’s feedback. When we are looking at individual patients who had samples at all three time points (Baseline, week 5, and week 12) we see a return to near baseline levels in two-thirds of this subset. However, in our conclusion of “After CRT, the gut microbiome’s diversity tends to return to baseline levels, but its structure and composition remain significantly altered” we are referring to the results of the full analysis that shows a non-significant difference between baseline and week 12 (Table 1). We recognize that this might cause confusion and therefore have removed the following sentence entirely from the manuscript: “Overall, two-thirds of patients’ diversity indices returned to near-baseline levels.”

- We have also included the duration of follow up as to not include any misleading conclusions. We have edited the conclusion in the abstract to read “After CRT, the diversity of the gut microbiomes in this population tended to return to baseline levels by the 12 week follow-up period, but structure and composition remained significantly altered.”

- Edits to the manuscript can be found on pages 3-4, lines 30-32; page 13, lines 214-215. 

REVIEWER 1, COMMENT 2 (Abstract) 

I agree with the following "These changes should be considered when designing studies to analyze the gut microbiome" but suspect that these temporal changes need to be considered for any study evaluating the gut microbiome in patient receiving chemoradiation and are not specific to only predictive/prognostic marker studies. The "predictive or prognostic biomarker . . ." phrase seems to come of out nowhere. In the discussion, "treatment response and toxicity" are mentioned.

- We thank the reviewer for their excellent suggestion. We agree with the reviewer and have revised the statement. The statement now reads “These changes should be considered when designing studies to analyze the gut microbiome in patients who receive pelvic CRT for gynecologic cancers.”

- Edits to the manuscript can be found on page 4, lines 34-36. 

REVIEWER 1, COMMENT 1 (Results) 

Please include information regarding extended field radiation to the PA nodes.

- We thank the reviewer for their feedback. We performed subset analyses of patients who received para-aortic nodal radiation vs. those who did not and observed no differences in these patients in our study in either alpha or beta diversity. However, it is impossible to make a conclusion regarding this given the small patient numbers. Instead, we have now added the following statement regarding extended field radiation to the results section. “Only 4 patients received extended field radiation to the Para-Aortic (PA) nodes.”

- Edits to the manuscript can be found on page 10, line 161. 

REVIEWER 1, COMMENT 2 (Results) 

Provide additional information regarding the reason for antibiotic use.

- We appreciate the reviewer’s comment. We have added “Reasons for antibiotic use among our patients include urinary tract infections (UTI), skin superinfections, and use of a urinary foley catheter.” to the results section. 

- Edits to the manuscript can be found on page 10, lines 165-167. 

REVIEWER 1, COMMENT 3 (Results) 

The authors "found no significant differences in diversity metrics between patients who took antibiotics prior to each time point and those who didn’t." However, with such a high number of patients receiving antibiotics at various time points there most likely was not adequate power to assess this question. Only 16 patients did not take antibiotics.

- We appreciate the reviewer’s comment. We recognize that our study was not powered to identify a difference in the microbiome between those who took antibiotics and those who did not. However, we conducted this analysis for completion and have reviewed our conclusions to make it clear that these results were those that we observed among our patient population. The statement now reads “We found no significant differences in diversity metrics between patients in our study population who took antibiotics prior to each time point and those who didn’t.”

- We have also added to the limitations sections to reflect the high use of antibiotics in our study. “Patients undergoing radiation for cervical cancer are at a high risk of urinary tract infections and skin infections. Many patients have advanced tumors which require nephrostomy tubes and/or indwelling urinary catheters which put them at a higher risk of infection. Thus, due to the high usage of antibiotics among our patients we did not have adequate power to identify differences in the gut microbiome between those who took antibiotics and those who did not. However, our results highlight the importance of continued research on this understudied topic.” was added to the limitation section 

- Edits to the manuscript can be found on page 11, lines 197-199; page 23, lines 423-429. 

REVIEWER 1, COMMENT 1 (Discussion) 

Further explain the high attrition rate - how does this compare to other studies of the gut microbiome?

-We thank the reviewer for their feedback. Our study compares relatively well to other studies that have prospectively sampled the gut microbiome in papers. Work done by Claesson et al. studying the variations in the microbiome of elderly adults has started with 161 samples at baseline and only had 23 at their 3 month follow up. Moreover, a review article discussing the different aspects of studying the microbiome mentions the challenges associated with prospective trials studying the microbiome2. We have added to the limitation section the possible reasons for our attrition rate. “The only requirement to be included in this study was the collection of a baseline sample which might explain this high attrition rate. Patients were offered the opportunity to give samples at each follow-up time point but were not required to do so. Moreover, we were unable to collect samples at certain time points if patients were in pain.” has been added to the strengths and limitations section. 

- Edits to the manuscript can be found on page 23, lines 415-419. 

1. Claesson MJ, Cusack S, O’Sullivan O, et al. Composition, variability, and temporal stability of the intestinal microbiota of the elderly. PNAS. 2011;108(Supplement 1):4586-4591. doi:10.1073/pnas.1000097107

2. Gerber GK. The dynamic microbiome. FEBS Letters. 2014;588(22):4131-4139. doi:10.1016/j.febslet.2014.02.037

REVIEWER 1, COMMENT 2 (Discussion) 

Provide explanation for high antibiotic use.

- We appreciate the reviewer’s comment. We have added the following to the discussion “Patients undergoing radiation for cervical cancer are at a high risk of urinary tract infections and skin infections. Many patients have advanced tumors which require nephrostomy tubes and/or indwelling urinary catheters which put them at a higher risk of infection.” 

- Edits to the manuscript can be found on page 23, lines 423- 426

REVIEWER 1, COMMENT 3 (Discussion) 

Lines 308-310 - why do the authors believe a shift in gut microbiome with CRT will influence outcome? Add reference. If no reference revise this sentence 

- We thank the reviewer for their feedback. We have actually shown that the gut microbiome does impact survival outcomes in the same cohort of patients. We have now added the reference to our paper: “Sims TT, Alam MBE, Karpinets TV, et al. Gut microbiome diversity is an independent predictor of survival in cervical cancer patients receiving chemoradiation. medRxiv. Published online May 22, 2020:2020.05.18.20105809. doi:10.1101/2020.05.18.20105809”

- Edits to the manuscript can be found on page 19, lines 323-325. 

REVIEWER 1, COMMENT 4 (Discussion) 

Line 337 How do you know Bacteroidetes, remained permanently altered when you only have short term follow up? "permanently altered" needs to be revised to reflect the period of time in your study

- We thank the reviewer for their comment. We have revised the manuscript to not provide any misleading conclusions with respect to the follow up period. We have revised the statement and added the duration of follow up: “Other species, namely Bacteroidetes, remained altered during our 12 week follow up period”. 

- Edits to the manuscript can be found on page 20, lines 352-353

REVIEWER 1, COMMENT 5 (Discussion) 

Line 359-361 - The authors state "Up to 20% of patients who have had CRT have long-term diarrhea after CRT, and it is possible these changes are related to lack of re-establishment of gut microbiome." Could it also be possible that radiation therapy may effect stool frequency and bile salt absorption and that leads to the change in gut microbiota? Your study provides important preliminary information, however, there is still so much to learn about the long-term intestinal effects from radiation therapy and the relationship to the gut microbiome. It would be helpful to add information regarding long term damage to the intestinal mucosa from radiation if available in the literature.

We appreciate the reviewer’s comment. While this is a very important topic, it is still under study. Our collaborative group has previously shown that increased acute radiation toxicity is associated with an alteredgut microbiome, which may be an association rather than causation. It is possible that the mucosa is damaged which could in turn affect the gut microbiome and cause a decrease in diversity. This topic remains under investigation since the science mainly shows an association between the two rather than a causation in any one direction. We have added “We have previously shown that increased short term radiation toxicity is associated with a decreased gut microbiome1. Many studies have sought to examine this relationship between the intestinal effects of radiotherapy and the gut microbiome but have mostly established an association rather than a causation in any one direction 2-3.” to the discussion section.

1. Mitra A, Grossman Biegert GW, Delgado AY, et al. Microbial Diversity and Composition Is Associated with Patient-Reported Toxicity during Chemoradiation Therapy for Cervical Cancer. International Journal of Radiation Oncology*Biology*Physics. 2020;107(1):163-171. doi:10.1016/j.ijrobp.2019.12.040

2. Ferreira MR, Andreyev HJN, Mohammed K, et al. Microbiota- and Radiotherapy-Induced Gastrointestinal Side-Effects (MARS) Study: A Large Pilot Study of the Microbiome in Acute and Late-Radiation Enteropathy. Clin Cancer Res. 2019;25(21):6487-6500. doi:10.1158/1078-0432.CCR-19-0960 

3. Kumagai T, Rahman F, Smith AM. The Microbiome and Radiation Induced-Bowel Injury: Evidence for Potential Mechanistic Role in Disease Pathogenesis. Nutrients. 2018;10(10):1405. doi:10.3390/nu10101405

- Edits to the manuscript can be found on page 21, lines 376-380.

REVIEWER 1, COMMENT 6 (Discussion) 

Lines 362-364 - revise this sentences to reflect limited power in your study to determine association between antibiotics and the change in the gut microbiome. If you do have adequate power then please add this information

- We thank the reviewer for their comment. We have revised the sentence and now reads as follows “This study was not powered to detect effects of antibiotics on the gut microbiome, but we did not find an association between the administration of antibiotics for treatment-related toxicities and alteration of the gut microbiome in this patient population. ”

- Edits to the manuscript can be found on page 21, line 383-386. 

REVIEWER 1, COMMENT 7 (Discussion) 

Lines 366-368 It is not clear how your study showed that intentionally modifying the gut microbiota could be used to reduce toxicity. Your results indicate a temporal change in microbiota composition in patients undergoing chemoradiation. Materials and methods don't describe an intervention to modify the gut microbiome; and results don't include toxicity data.

- We appreciate the reviewer’s comment. The aim of that statement was mainly to show that modifying the gut microbiome to accumulated species that tolerate treatment could be used to reduce CRT induced toxicity. We see how that statement might be misleading and so we have edited it to: “Modifying the gut microbiome to accumulate CRT-tolerant species could potentially be used to reduce treatment toxicity.”

- Edits to the manuscript can be found on page 22, lines 388-389

---

## [Decision Letter · Decision Letter 1]

2 Feb 2021

PONE-D-20-16939R1

Adaptive changes in the gut microbiome during standard-of-care chemoradiotherapy for gynecologic cancers

PLOS ONE

Dear Dr. Colbert,

Thank you for submitting your manuscript to PLOS ONE. After careful consideration, we feel that it has merit but does not fully meet PLOS ONE’s publication criteria as it currently stands. Therefore, we invite you to submit a revised version of the manuscript that addresses the points raised during the review process.

Your revised manscript was reviewed by two new reviewers who both felt the comments from the first round of review were well addressed. Minor additional revisions are suggested based on the comments of Reviewer 3. In addition raw data should be deposited in a public repository (e.g., SRA).

We look forward to receiving your revised manuscript.

Kind regards,

Christopher Staley, Ph.D.

Academic Editor

PLOS ONE

Reviewers' comments:

Reviewer's Responses to Questions

**Comments to the Author**

1. If the authors have adequately addressed your comments raised in a previous round of review and you feel that this manuscript is now acceptable for publication, you may indicate that here to bypass the “Comments to the Author” section, enter your conflict of interest statement in the “Confidential to Editor” section, and submit your "Accept" recommendation.

Reviewer #2: All comments have been addressed

Reviewer #3: (No Response)

2. Is the manuscript technically sound, and do the data support the conclusions?

Reviewer #2: Yes

Reviewer #3: Yes

3. Has the statistical analysis been performed appropriately and rigorously? 

Reviewer #2: Yes

Reviewer #3: Yes

4. Have the authors made all data underlying the findings in their manuscript fully available?

Reviewer #2: Yes

Reviewer #3: No

5. Is the manuscript presented in an intelligible fashion and written in standard English?

Reviewer #2: Yes

Reviewer #3: Yes

6. Review Comments to the Author

Reviewer #2: All of the questions and comments have eenn addressed satisfactorioy by the authors.

This manuscript is timely and appropriate.

Reviewer #3: This is a short-term prospective study of changes in the rectal microbiome due to chemotherapy and radiation therapy in adult women with (primarily) cervical cancer. The study design, specimen collection, and techniques used in assessing the rectal microbiome were straightforward. Interpretation of findings, however, were complicated by the collection of relatively few samples at the end of the study (only 16 samples at week 12 from a total of 58 participants at baseline) and a large percentage of participants who received antibiotic therapy during the study. The authors’ responses to reviewer comments were appropriate and enhanced the clarity of the manuscript. A few additional comments are provided for the authors’ consideration.

Consider adding “prospective” to the title so as to better frame your study design for readers. For example, the title might read, “Prospective study of adaptive changes in the gut microbiome…”

All eligible patients who were seen in clinic were given the opportunity to participate in the study. 58 individuals consented. What was the total number of patients who were approached? This should be stated in the Materials and Methods. Were there any significant differences between those who agreed to participate in the study and those who did not?

It is important to include in the abstract the number of subjects who received antibiotics during the study period. This represents a major confounder and should be rightfully mentioned in the summary.

Line 125, “For each patient who provided samples at all five time points (n=17)…” – only 16 samples were counted in Table S2. Please reconcile.

Lines 127-128, “for patients who did not provide samples at all time points (n=58)” – should not this be 16 (or 17)? Please clarify.

Please provide the range and average sequence sampling depth for all samples included in the study. Sufficient sampling depth is needed so that minor OTUs are not overlooked. The authors should assure that OTUs were assessed with >95% confidence at, for example, a relative abundance of >0.1% or >0.01%. Sufficient sequence sampling depth is needed to minimize artificial skewing of diversity assessments.

So few samples were collected at week 12 (16 [17%] out of 58 initially enrolled subjects) that any comparison to baseline samples is difficult to interpret (lines 199 – 215). It cannot be known how biases may have been introduced by dropouts that subsequently affected comparisons. This limitation was not adequately discussed. Frankly, the stark attrition at week 12 suggested to this reviewer that results for week 12 as shown in Table 1 and Fig. 2, if included at all, should be relegated to supplemental data, as was the case with supplemental Fig. 3, and caveats included in the text about the limitations to any interpretation.

7. PLOS authors have the option to publish the peer review history of their article (what does this mean?). If published, this will include your full peer review and any attached files.

Reviewer #2: No

Reviewer #3: **Yes: **Mark M Huycke

---

## [Author Response · Author response to Decision Letter 1]

15 Feb 2021

EDITOR, COMMENT 1

Raw data should be deposited in a public repository (e.g., SRA).

Response:

We have deposited our raw data in the Sequence Read Archive (SRA). The project accession number is PRJNA685389

REVIEWER 3, COMMENT 1 (Title)

Consider adding “prospective” to the title so as to better frame your study design for readers. For example, the title might read, “Prospective study of adaptive changes in the gut microbiome…”

Response:

We appreciate the reviewer’s suggestion. We have changed the title to “A prospective study of the adaptive changes in the gut microbiome during standard-of-care chemoradiotherapy for gynecologic cancers”

REVIEWER 3, COMMENT 2 (Methods) 

All eligible patients who were seen in clinic were given the opportunity to participate in the study. 58 individuals consented. What was the total number of patients who were approached? This should be stated in the Materials and Methods. Were there any significant differences between those who agreed to participate in the study and those who did not?

Response:

We appreciate the reviewer’s comment. In our study, every eligible patient was approached to be included in our study, however some patients refused to participate since they did not want additional tissue samples taken. We are not sure of the exact number of patients approached during this study period, but our clinic sees on average 75 patients per year. We have added “Our clinic sees an average of 75 patients per year.” in the methods section.

-Edits to the manuscript can be found on page 6, lines 83-84 

REVIEWER 3, COMMENT 3 (Abstract) 

It is important to include in the abstract the number of subjects who received antibiotics during the study period. This represents a major confounder and should be rightfully mentioned in the summary.

Response:

We thank the reviewer for their suggestion. We have included the number of subjects receiving antibiotics during the study in the abstract. “42 of these patients received antibiotics during the study period.”

- Edits can be found on page 3, lines 17-18. 

REVIEWER 3, COMMENT 4 (Methods) 

Line 125, “For each patient who provided samples at all five time points (n=17)…” – only 16 samples were counted in Table S2. Please reconcile.

Response:

We appreciate the reviewer’s comment. For this specific analysis used for figure 2, we used data from patients who had samples at all three timepoints (baseline, week 5, and week 12). This seems to have been a typo, thank you for your attention to detail. We have revised this sentence and now reads “for each patient who provided samples at baseline, week 5, and week 12 (n=11), we tracked the changes in diversity over time to determine the number of patients for whom gut diversity returned to baseline levels.” 

- Edits to the manuscript can be found on page 8, lines 127-130.

REVIEWER 3, COMMENT 5 (Methods) 

Lines 127-128, “for patients who did not provide samples at all time points (n=58)” – should not this be 16 (or 17)? Please clarify.

Response:

We thank the reviewer for their comment. We built a linear mixed model accounting for patients who had missing data for one or more timepoints. In our study, we had 53 patients with at least one missing time point and so we have corrected the sentence and now reads “For patients who did not provide samples at all time points (n=53), we built a linear mixed model for time as a predictor of diversity” 

- Edits to the manuscript can be found on page 8, line 133 

REVIEWER 3, COMMENT 6 (Methods) 

Please provide the range and average sequence sampling depth for all samples included in the study. Sufficient sampling depth is needed so that minor OTUs are not overlooked. The authors should assure that OTUs were assessed with >95% confidence at, for example, a relative abundance of >0.1% or >0.01%. Sufficient sequence sampling depth is needed to minimize artificial skewing of diversity assessments.

Response:

We thank the reviewer for their suggestion. We have added the range and mean sequence sampling depth for all samples included in the study in a new Supplemental Table 1. We have also included the following statement in the methods section: “The range and average sequence sampling depth for each sample is provided in S1 Table.” We have also mentioned in the text that the OTU assignment was performed with 97% sequence similarity.

- Edits to the manuscript can be found on page 8, lines 117-118; and pages 30-31, lines 531-533

REVIEWER 3, COMMENT 7 (Discussion) 

So few samples were collected at week 12 (16 [17%] out of 58 initially enrolled subjects) that any comparison to baseline samples is difficult to interpret (lines 199 – 215). It cannot be known how biases may have been introduced by dropouts that subsequently affected comparisons. This limitation was not adequately discussed. Frankly, the stark attrition at week 12 suggested to this reviewer that results for week 12 as shown in Table 1 and Fig. 2, if included at all, should be relegated to supplemental data, as was the case with supplemental Fig. 3, and caveats included in the text about the limitations to any interpretation.

Response:

We appreciate the reviewer’s comment. We have tried to be very careful with our conclusions regarding data at week 12. Although a high attrition rate in studying is of valuable concern, our study is a longitudinal one and so patients were compared to themselves using paired statistical tests. We have removed the results of week 12 in Table one and have included the following sentence to the results section. “Diversity at week 12 was also compared to baseline, however we have not included these results in Table 1 due to a low patient number at this time point.”. We would prefer to keep Figure 2 as a visualization but have taken care to not present any statistical tests. The aim of the figure is to visualize and trace the diversity over time in patients who had provided samples for all three time points. We have clarified this point in the figure legend and text.

-Edits to the manuscript can be found on page 11, lines 194-195

---

## [Editor Report · Decision Letter 2]

17 Feb 2021

A prospective study of the adaptive changes in the gut microbiome during standard-of-care chemoradiotherapy for gynecologic cancers

PONE-D-20-16939R2

Dear Dr. Colbert,

We’re pleased to inform you that your manuscript has been judged scientifically suitable for publication and will be formally accepted for publication once it meets all outstanding technical requirements.

Kind regards,

Christopher Staley, Ph.D.

Academic Editor

PLOS ONE
---

## [Editor Report · Acceptance letter]

19 Feb 2021

PONE-D-20-16939R2 

A prospective study of the adaptive changes in the gut microbiome during standard-of-care chemoradiotherapy for gynecologic cancers 

Dear Dr. Colbert:

I'm pleased to inform you that your manuscript has been deemed suitable for publication in PLOS ONE. Congratulations! Your manuscript is now with our production department. 

Kind regards, 

on behalf of

Dr. Christopher Staley 

Academic Editor

PLOS ONE